# Biomathematical enzyme kinetics model of prebiotic autocatalytic RNA networks: degenerating parasite-specific hyperparasite catalysts confer parasite resistance and herald the birth of molecular immunity

**Magnus Pirovino[1], Christian Iseli [2], Joseph A. Curran[3], Bernard Conrad [4]***

**1** OPIRO Consulting Ltd., Triesen, Principality of Liechtenstein, **2** Bioinformatics Competence Center, EPFL and Unil, Lausanne, Switzerland, **3** Department of Microbiology and Molecular Medicine, and Institute of Genetics and Genomics of Geneva (iGE3), Geneva University, Genève, Switzerland, **4** Genesupport, Avenue de Sévelin 18, Lausanne, Switzerland

* bernard.conrad@genesupport.ch

## Abstract

Catalysis and specifically autocatalysis are the quintessential building blocks of life. Yet, although autocatalytic networks are necessary, they are not sufficient for the emergence of life-like properties, such as replication and adaptation. The ultimate and potentially fatal threat faced by molecular replicators is parasitism; if the polymerase error rate exceeds a critical threshold, even the fittest molecular species will disappear. Here we have developed an autocatalytic RNA early life mathematical network model based on enzyme kinetics, specifically the steady-state approximation. We confirm previous models showing that these second-order autocatalytic cycles are sustainable, provided there is a sufficient nucleotide pool. However, molecular parasites become untenable unless they sequentially degenerate to hyperparasites (i.e. parasites of parasites). Parasite resistance–a parasite-specific host response decreasing parasite fitness–is acquired gradually, and eventually involves an increased binding affinity of hyperparasites for parasites. Our model is supported at three levels; firstly, ribozyme polymerases display Michaelis-Menten saturation kinetics and comply with the steady-state approximation. Secondly, ribozyme polymerases are capable of sustainable auto-amplification and of surmounting the fatal error threshold. Thirdly, with growing sequence divergence of host and parasite catalysts, the probability of self-binding is expected to increase and the trend towards cross-reactivity to diminish. Our model predicts that primordial host-RNA populations evolved via an arms race towards a host-parasite-hyperparasite catalyst trio that conferred parasite resistance within an RNA replicator niche. While molecular parasites have traditionally been viewed as a nuisance, our model argues for their integration into the host habitat rather than their separation. It adds another mechanism–with biochemical precision–by which parasitism can be tamed and offers an attractive explanation for the universal coexistence of catalyst trios within prokaryotes and the virosphere, heralding the birth of a primitive molecular immunity.

**Data Availability Statement:** The data, including scripts, is fully made available at the following sites (https://github.com/BICC-UNIL-EPFL/biomathematical-enzyme-kinetics-model [github.com], and the DOI: https://doi.org/10.5281/zenodo.11060828 [doi.org].

**Funding:** The author(s) received no specific funding for this work.

**Competing interests:** The authors have declared that no competing interests exist.

## Author summary

The quintessential components of life comprise a potent mixture of naturally occurring, but improbable chemical reactions (catalysis), and the arrangement of such accelerated chemical reactions into closed loops (autocatalytic sets). This is required but is not sufficient for such networks to self-propagate (amplification of the information carrier = host polymerization) and adapt (Darwinian evolution). As soon as self-propagation is attained, the next hurdle is parasitism. This typically involves shorter molecules (the products of replicative errors) that hitchhike the replicative potential of the host. They will invariably outcompete the regular amplification process, unless a solution is found. We have addressed this problem using a new model based on the mathematics of catalysis. It confirms previous studies demonstrating that autocatalytic sets become self-sustaining, assuming that a sufficient pool of molecular building blocks is available. However, molecular parasitism is pervasive and potentially fatal for both host and parasite. In our model, we allow these parasites to degenerate in a controlled fashion, giving rise to parasites of parasites (hyperparasites). As long as these hyperparasites acquire binding specificity for parasites, an attenuation of parasitism is observed. These parasite-hyperparasite cycles stabilize the host cycle, explaining why they are conserved, and why they are the likely reason behind the observation that all cellular hosts are associated with parasites (e.g. bacteria) and hyperparasites (e.g. viruses) across all kingdoms of life. Moreover, it provides a novel solution to the usually intractable problem of parasitism.

## Introduction

The multistep transition process from inanimate to living systems has been dated to >3.9 Ga [1]. A non-exhaustive list of thresholds that needed to be progressively surmounted during this process has been drawn, comprising, among others, chirality symmetry breaking, spontaneous polymerization, and self-assembly of compartments. The catalytic closure threshold and Eigen's error threshold have received particular attention [2]. The latter posits that if the replication error exceeds a critical threshold value even the fittest replicators will not survive [3].

One central pillar in the theory of how early life could have materialized pertains to the "RNA world". This stipulates that the initial informational polymer was RNA, and that the same molecule was also the fundamental catalyst [4]. The underlying concept of autocatalytic sets was laid out more than half a century ago [3,5]. It was subsequently refined [6] and validated in a proof of principle experiment showing that RNA fragments could build self-replicating ribozymes from individual RNA fragments via cooperative catalytic cycles; networks with three members showed strong, highly cooperative growth [7]. In parallel, an increasingly efficient number of RNA polymerase ribozymes were generated by *in vitro* evolution. These exhibited high levels of processivity [8], and displayed Michaelis-Menten saturation kinetics [9]. *In fine*, one RNA polymerase ribozyme surpassed Eigen's critical error threshold, and successfully accomplished several rounds of cyclic self-replication [10,11]. Collectively, both theoretical considerations and experimental evidence support the view of a gradual evolution towards RNA self-catalysis. This started with self-assembly, followed by the formation of autocatalytic networks using self-assembled substrates. From this emerged different ribozymes (recombinases, ligases) and template based ligation, leading to a template based, self-replicase catalysed assembly of monomers [12].

The ribozyme catalysed RNA-polymerization process is even more error prone [4,8–11] than the proteinaceous RNA polymerases that notoriously lack proofreading activity [13]. This

generates variant sequence space mutants called quasispecies [14,15]. This variant sequence-space intrinsically implies the obligate presence and evolutionary persistence of molecular parasites, as observed in all life forms and can be argued essentially for thermodynamic reasons [16,17]. Two main mechanisms have been proposed to confer parasite resistance to a given habitat, namely spatial self-organization [18–24], and physical compartmentalisation most probably via lipid bilayers, and the construction of protocells [25–27]. A direct comparison indicates that the survival of RNA replicators most likely occurred in protocells [28].

Since catalysis and autocatalysis are central to life, from a mathematical perspective the Michaelis-Menten reaction scheme appears pertinent [29], and implicitly the quasi steady-state assumption [30]. These considerations lay the ground for modelling catalysis, autocatalysis and the interactions within primordial, template-based ecological RNA communities using a single biomathematical tool derived from enzyme kinetics. We model here, based on a previously established tripartite population scheme [31], how molecular parasite resistance and a primordial molecular antiparasite immunity gradually emerged from progressively degenerating hyperparasite catalysts acquiring parasite-specificity. All important terms, model parameters and notations used are defined in the Text Box (S1 Definitions of terms and annotations).

## Results

Enzyme kinetics lies at the centre of the living world, and the intermediate complex lies at the centre of the centre. Consequently, in developing our current model we have used equations derived from enzyme kinetics, specifically the steady-state approximation by Briggs and Haldane [32]. Starting with a ribozyme polymerase exhibiting Michaelis-Menten saturation kinetics capable of beating the error-threshold [9–11], we demonstrate how template-based RNA autocatalytic cycles progressively degenerate to create successively parasite and hyperparasite cycles. Metabolic autocatalytic networks, arguably arising spontaneously and predating template-based polymerization [33], are assumed to be present and to provide sufficient energy and polymer precursors [23,31]. The previously established tripartite habitat framework, featuring a template-based RNA host and a molecular parasite that is itself parasitized (hyperparasite catalysts) [31] is instrumental for homeostasis and the emergence of molecular parasite resistance. It hinges upon hyperparasite catalysts acquiring parasite-specificity.

We provide mathematical models that describe the time evolution of autocatalytic RNA networks (Figs 1–4). The main goal of these models is to investigate the general conditions under which the stability of such networks could occur. The physical/chemical dimensions of the relevant input and output values (e.g. $t$ = time, $k_{cat}$, $k_{on}$, $k_{off}$, $K_M$, RNA-concentration values, etc.) are assumed to be dimensionless. However, when available numerical values have been employed. Nevertheless, these values were not validated with respect to a specific experimental system.

### Biomathematical Michaelis-Menten kinetics and the Briggs-Haldane steady-state approximation for autocatalytic RNA networks

It is assumed that the ribozyme polymerase represents the positive RNA strand $R_1$ with kinetic parameters capable of cyclic auto-amplification that surmount the error threshold (e.g. a $k_{cat}$ of 0.1–1.0 min$^{-1}$ and a $K_M$ akin or superior to 0.1–1.0 μM) [9–11]. Its complementary negative RNA strand $R_2$ is considered to be catalytically inert (uniquely a template). A complete auto-catalytic replication cycle comprises the initial production of $R_2$ by two $R_1$ ribozyme molecules, and the subsequent reproduction of $R_1$ by $R_1$ using template $R_2$ (Model 1, formalism described in *Methods or Models*, $R_1 \Rightarrow (R_1\text{-}R_1) \rightarrow^c R_2 \Rightarrow (R_1\text{-}R_2) \rightarrow^c R_1 (1.0)$, see also Figs 1–4 for scheme). Fig 5 shows a graphical representation of this model assuming realistic starting kinetics and

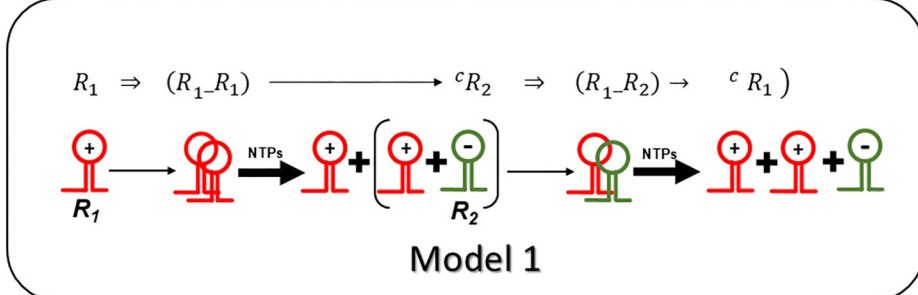

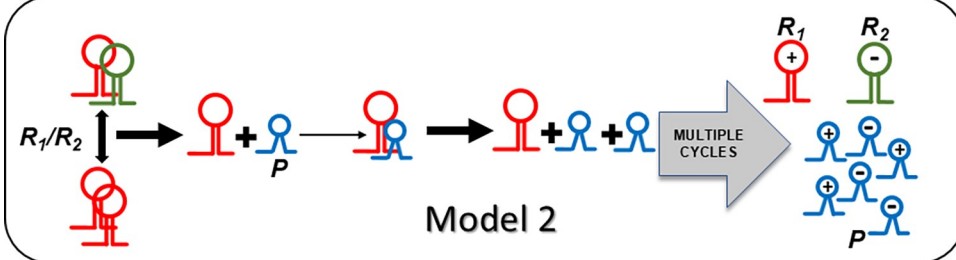

**Fig 1. The reaction schemes for the main models 1 and 2.** Fig 1 shows Model 1 (top panel) featuring a host cycle based on a second-order catalysis scheme. A negative-stranded ribozyme $R_2$ (-) produced by two positive-stranded $R_1$ molecules (+) serves as template, enabling efficient amplification of the host $R_1$ species, provided sufficient ribonucleoside 5' triphosphate (NTPs) building blocks are present. The lower panel displays Model 2 illustrating the emergence and persistence of molecular parasites $P$. $P$ molecules typically are smaller with shorter generation times and better templates then $R_1$ and $R_2$, overwhelming the habitat.

habitat restriction parameters. The model parameters are described in S1 Definition of terms and annotations. As can be seen, the abundance of these two molecular species increases rapidly and becomes stable, supporting previous work showing the viability of ribozyme polymerases, assuming that sufficient energy in the form of molecular building blocks is available [23].

## Molecular parasitism in autocatalytic cycles

Molecular parasites are an inherent feature of the replication process. For a number of reasons, they are inseparably associated with all life forms [16,17], with the most important being the

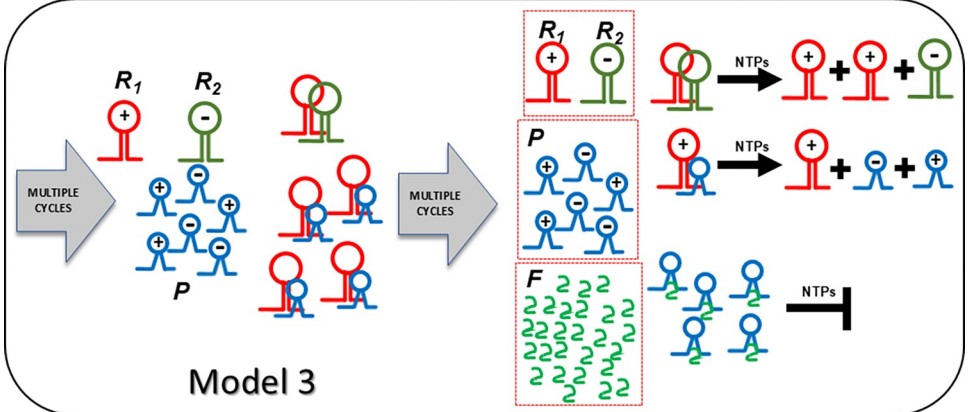

**Fig 2. The reaction scheme for the main model 3.** Fig 2 displays Model 3 representing the contemporaneous presence of hyperparasites $F$ and parasites $P$. $F$ recapitulates the features of $P$ (smaller, shorter generation time and better template then $P$, $R_1$ and $R_2$). Habitat homeostasis hinges upon $F$ binding to $P$ more specifically then to $R_1/R_2$.

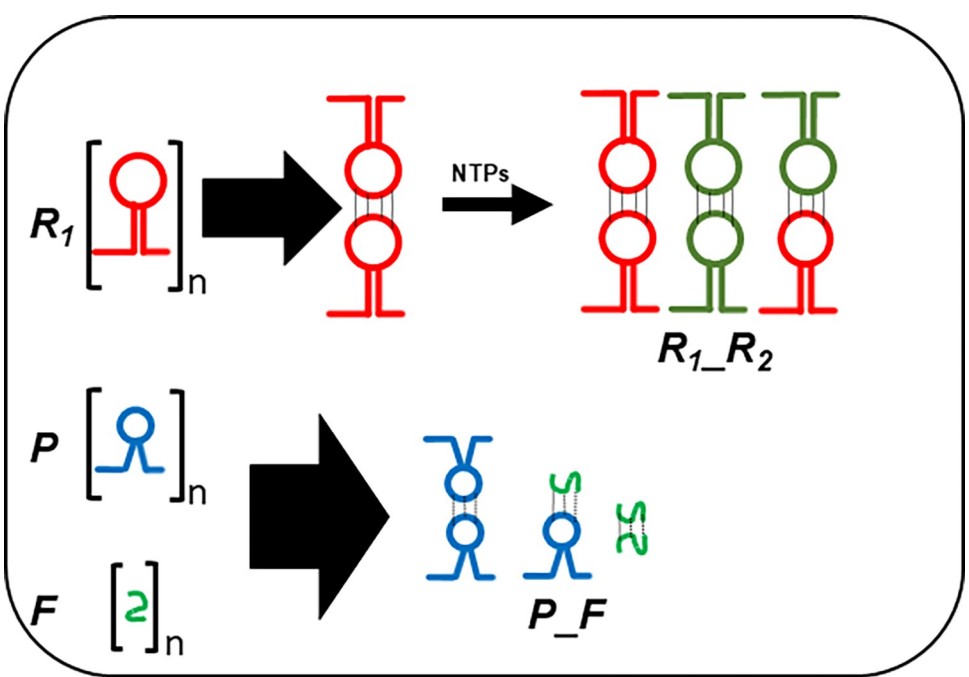

**Fig 3. The mechanistic reaction scheme for the main models.** Fig 3 shows mechanistically how parasites are tamed through increasing self-specific binding of host (upper row) and parasite/hyperparasites catalysts (lower rows), from left (lower affinity) to right of both arrows (higher affinity). Interactions are depicted as kissing loop interactions [46].

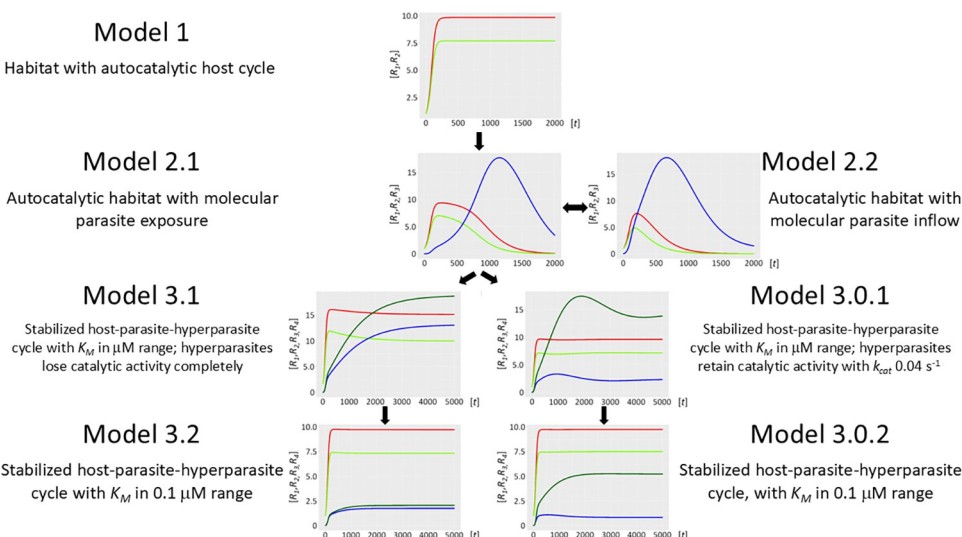

**Fig 4. Synopsis of the main models and of submodels.** The main Model 1 displays a habitat with a second-order autocatalytic ribozyme cycle. Model 2 comprises two variants, the central Model 2.1. is characterized by internal production of molecular parasites, while in Model 2.2 parasites flow in from a neighbouring habitat. The two submodels 3.1 and 3.0.1 of Model 3 debranch as alternatives from Model 2.1. Both imply stabilized host-parasite-hyperparasite cycles with $K_M$ in µM range. In Model 3.1 hyperparasites lose their catalytic activity instantly and completely, while in Model 3.0.1 (S1 Supporting Information) they initially retain a small but significant catalysis. Both models merge in Model 3.2 and Model 3.0.2 (S1 Supporting Information), which both provide definitive parasite taming by decreasing $K_M$ one order of magnitude.

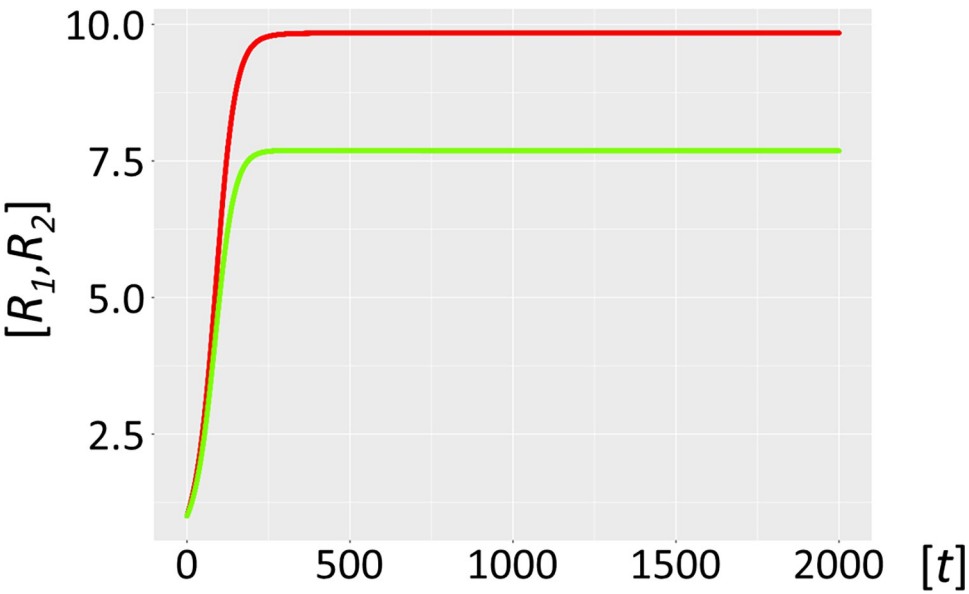

**Fig 5. Habitat with autocatalytic host cycle (Model 1).** Ribozyme concentrations of defined populations $R_1$ and $R_2$ [units on unspecified scale] are plotted against time $t$ [units on unspecified scale]. The host catalysts (ribozyme polymerase $R_1$, red, negative strand ribozyme $R_2$, light green) build a self-amplifying and sustained autocatalytic habitat, as long as sufficient nucleotides are provided.

accuracy-rate trade-off [34,35]. In general, enzymes operate orders of magnitude ($k_{cat}/K_M$ of $\sim 10^5 \, \mathrm{M^{-1}s^{-1}}$) below the diffusion rate ($10^8$–$10^9 \, \mathrm{M^{-1}s^{-1}}$). This implies promiscuous substrate binding, high catalytic activity, and as a secondary trade-off, a high mutation rate generating a large variance in the quasispecies: i.e. speed being more important than enzyme accuracy [36]. Selection favours parasite molecules of subgenomic size [37–39], since the smaller they are the shorter their replication time, and the more overwhelmingly abundant they will become [38]. In addition, host ribozyme polymerases $R_1$ amplifying parasites are unavailable for the synchronous host amplification cycle $R_1$-$R_2$. This is exactly what Model 2.1 shows (Fig 6, see also Figs 1–4 for scheme). It is merely a matter of time before the molecular parasites $R_3 = P$ outcompete the host molecules $R_1$ and $R_2$. Fig 7 shows a longer time scale of 5000 units, confirming the parasite-driven collapse of the habitat. Furthermore, if molecular parasites are diffusing in from a neighbouring habitat, the process is even more rapid (Model 2.2, Fig 8). Fig 9 corroborates this for a longer time scale of 5000 units. Therefore, as expected, high parasite loads overpower a typical habitat driven by efficient, but promiscuous ribozyme polymerases with $P$ populations. This conundrum begs a solution.

## The emergence of hyperparasites: stable autocatalytic cycles in the presence of molecular parasites

The host ribozyme polymerase $R_1$ is by design promiscuous ensuring amplification of both the host cycle $R_1$-$R_2$, and the very efficient amplification of the typically smaller $R_3 = P$ population (see above). Because of its high mutation rate, it will not only reproduce the original $P$ species, but will inevitably also generate $P$ mutants dubbed $F$, i.e. hyperparasites. Selection will preferentially amplify hyperparasite populations smaller than $P$ [37–39], (Model 3, formally described in *Methods or Models*, $R_1 \Rightarrow (R_1 P) \rightarrow^c F(1\&2 \rightarrow 3)$, see also Fig 2 for scheme). Short and loosely folded RNA sequences will be preferentially replicated (they are better templates), whereas longer and tightly folded ones may be better ribozymes and more resistant to

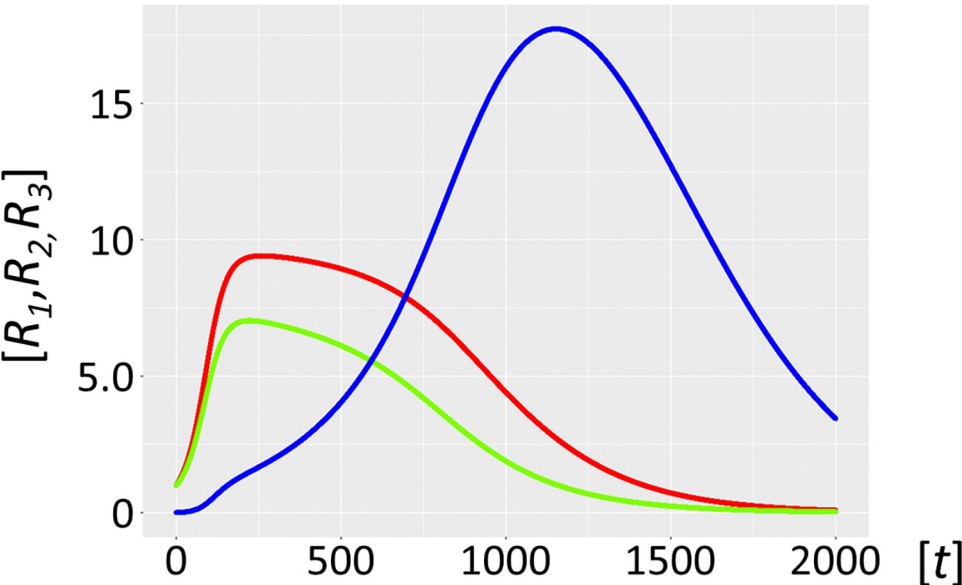

**Fig 6. Autocatalytic habitat with molecular parasite exposure (Model 2.1).** Ribozyme concentrations of defined populations $R_1$, $R_2$ and $R_3$ [units on unspecified scale] are plotted against time $t$ [units on unspecified scale]. The host catalysts (ribozyme polymerase $R_1$, red, negative strand ribozyme $R_2$, light green) are rapidly invaded by the parasite ribozyme species $R_3$ (blue), which is more efficiently amplified by the host ribozyme polymerase $R_1$. The parasites $R_3$ are mutants of the host cycle represented by $R_1$ and $R_2$, with mutation rate $\mu = 0.01$. Here we have no parasite inflow from neighbouring habitats, i.e. $\alpha = 0$.

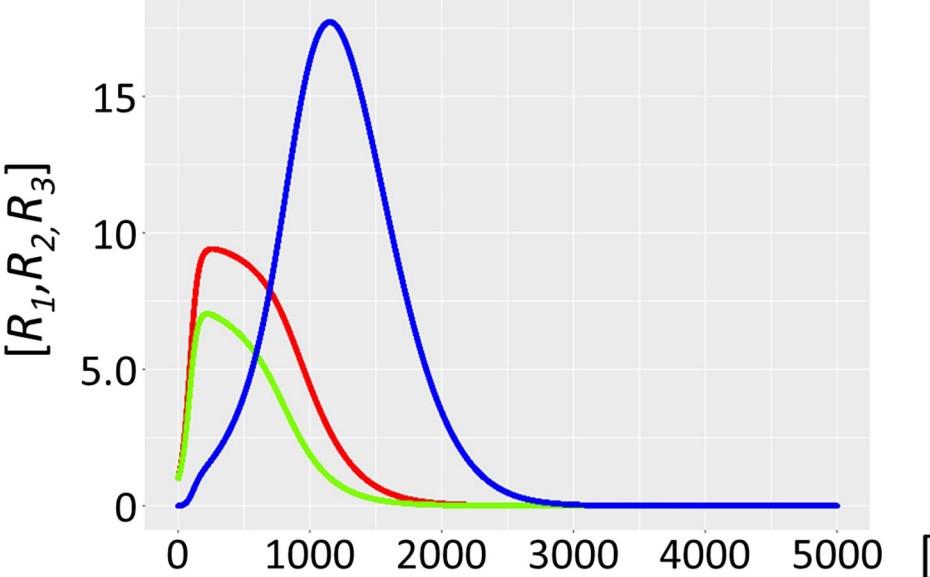

**Fig 7. Autocatalytic habitat with molecular parasite exposure (Model 2.1).** Ribozyme concentrations of defined populations $R_1$, $R_2$ and $R_3$ [units on unspecified scale] are plotted against time $t$ [units on unspecified scale]. The host catalysts (ribozyme polymerase $R_1$, red, negative strand ribozyme $R_2$, light green) are rapidly invaded by the parasite ribozyme species $R_3$ (blue), which is more efficiently amplified by the host ribozyme polymerase $R_1$. The parasites $R_3$ are mutants of the host cycle represented by $R_1$ and $R_2$, with mutation rate $\mu = 0.01$. Here we have no parasite inflow from neighbouring habitats, i.e. $\alpha = 0$. Fig 7 has identical parameters to Fig 6, except for a longer timescale of 5000 units.

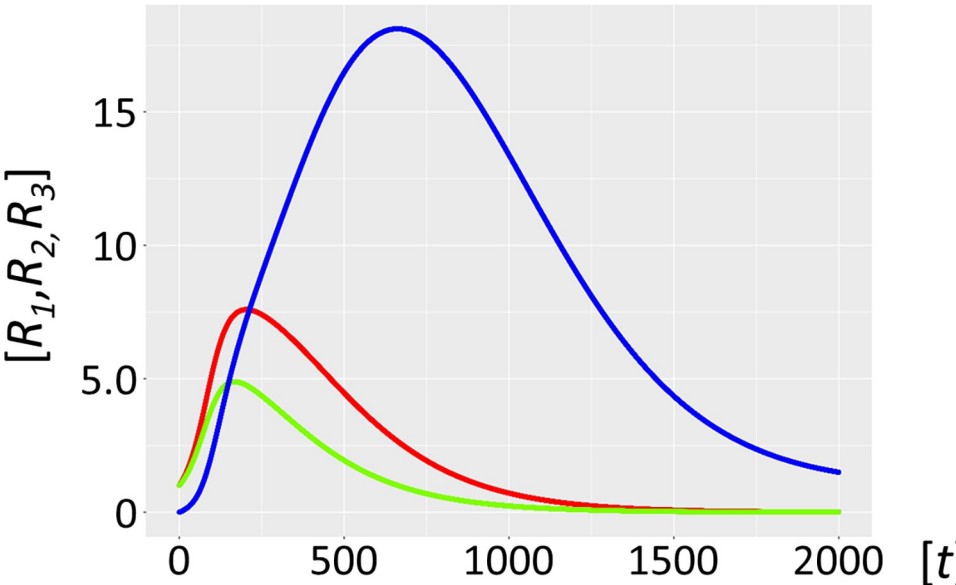

**Fig 8. Autocatalytic habitat with molecular parasite inflow (Model 2.2).** Ribozyme concentrations of defined populations $R_1$, $R_2$ and $R_3$ [units on unspecified scale] are plotted against time $t$ [units on unspecified scale]. If parasite ribozymes $R_3$ (blue) diffuse in from a neighbouring habitat, with parasite inflow per unit of time, $\alpha = 0.005$, the host catalysts (ribozyme polymerase $R_1$, red), and the negative strand ribozyme $R_2$ (light green) are even more rapidly outcompeted and disappear. Here the parasites $R_3$ **are** not mutants from the host cycle, represented by $R_1$ and $R_2$, i.e. mutation rate is $\mu = 0$, $R_3$ flow in from a neighbouring habitat.

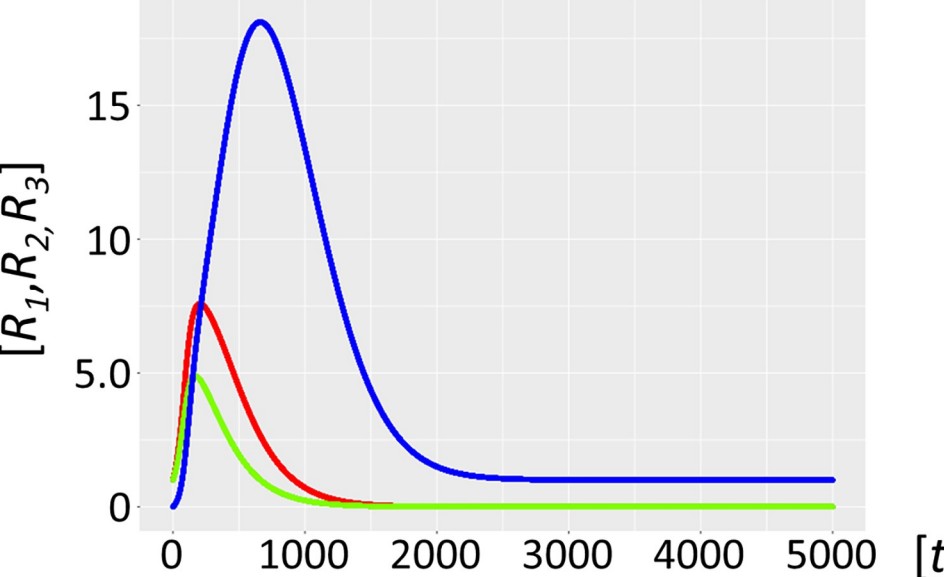

**Fig 9. Autocatalytic habitat with molecular parasite inflow (Model 2.2).** Ribozyme concentrations of defined populations $R_1$, $R_2$ and $R_3$ [units on unspecified scale] are plotted against time $t$ [units on unspecified scale]. If parasite ribozymes $R_3$ (blue) diffuse in from a neighbouring habitat, with parasite inflow per unit of time, $\alpha = 0.005$, the host catalysts (ribozyme polymerase $R_1$, red), and the negative strand ribozyme $R_2$ (light green) are even more rapidly outcompeted and disappear. Here the parasites $R_3$ **are** not mutants from the host cycle, represented by $R_1$ and $R_2$, i.e. mutation rate is $\mu = 0$, $R_3$ flow in from a neighbouring habitat. Fig 9 has identical parameters to Fig 8, except for a longer timescale of 5000 units.

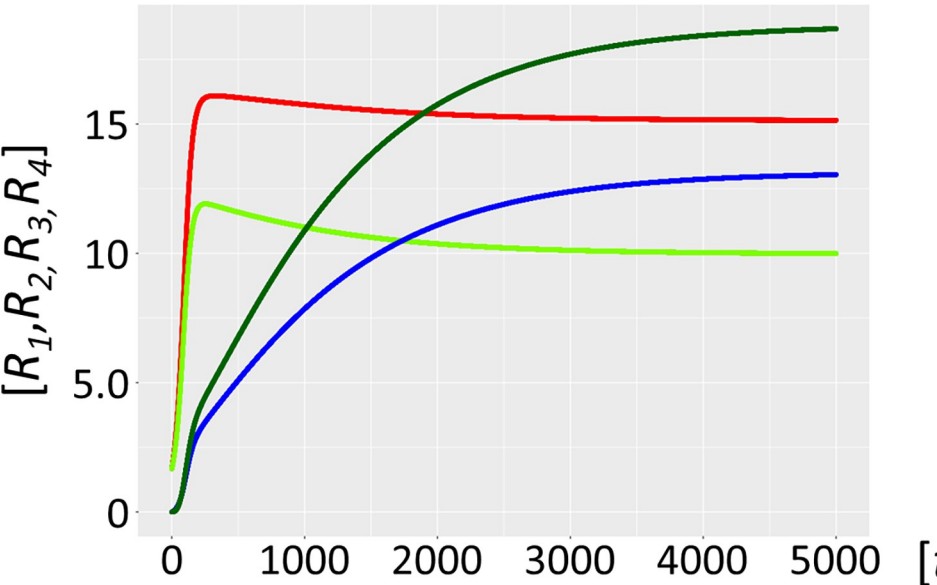

**Fig 10. Stabilized host-parasite-hyperparasite cycle with $K_M$ in μM range.** Ribozyme concentrations of defined populations $R_1$, $R_2$, $R_3$ and $R_4$ [units on unspecified scale] are plotted against time $t$ [units on unspecified scale]. A habitat with highly efficient catalysts (ribozyme polymerase $R_1$, red, negative strand ribozyme $R_2$, light green) becomes stabilized by the generation and persistence of hyperparasites (parasite of parasite) $F$ (dark green) that parasitize the parasites $R_3$ (blue). This already happens if intermediate complexes $P\_F$ and $F\_F$ have a $K_M$ in the μM range (Model 3.1). However, the hyperparasite (dark green) and parasite $R_3$ (blue) ribozymes remain prevalent under these conditions.

hydrolytic decay [40]. As a corollary, $F$ will be a highly efficient template, yet a poor or non-functional ribozyme capable of self-amplification. $R_1$ is already efficiently producing $P$ and $F$. However, this process is not subject to evolutionary pressure and is bound to degenerate $(F \Rightarrow (P\_F) \rightarrow^c nil, F \Rightarrow (F\_F) \rightarrow^c nil$ (3.0)). The hyperparasite will parasitize $P$ for several reasons; firstly, while bound in intermediate $P\_F$ complexes it is not available to $R_1$, which can again engage in the amplification of the host cycle $R_1$-$R_2$. Secondly, in order for hyperparasitism to be highly efficient, it is required that $F$ has a high specificity for $P$, i.e. in the presence of both $R_1$ and $P$ it binds mainly or exclusively to the latter. Indeed, this is precisely what the accuracy-rate trade-off predicts, with selectivity implying the ability to discriminate between two templates when both are present [35], whereas rate denotes speed [35,36]. Ground-state discrimination means that specificity is achieved mainly through substrate binding, which imposes strong accuracy/rate trade-offs [34,35]. Improvements in selectivity mediated by tighter cognate substrate binding invariably leads to lower catalytic efficiency (parallel decreases in the constants for the cognate substrates $K_M^{cog}$ and $k_{cat}^{cog}$ [34,35]). This is precisely what the model stipulates; $F$ has parasite- and hyperparasite-selectivity, which comes at the cost of reduced catalytic activity as would be predicted for a smaller molecular species. As shown in Fig 10 (Model 3.1), and assuming the standard parameters as applied in Model 1, with similar $K_M$ values (in the μM range) for both the intermediate complexes ($P\_F$ and $F\_F$) and $R_1\_R_2$, hyperparasitism *per se* sustainably restores the abundance of the host cycle species $R_1$ and $R_2$. Still, the hyperparasite $F$ is the most abundant species in the habitat, and the level of $P$ remains important.

However, increasing the binding affinity of $F$ for itself and $P$ by one order of magnitude ($K_M$ values for $P\_F$ and $F\_F$ now in the 0.1 μM range, Model 3.2, Fig 11, see also Fig 4 for scheme) fully restores the abundance of the host cycle RNA species $R_1$ and $R_2$, at levels

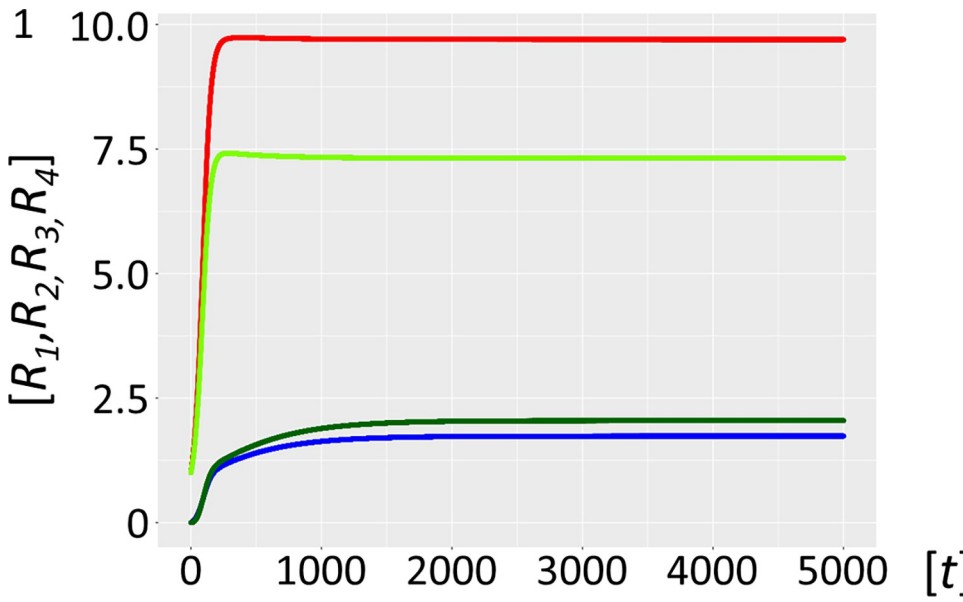

**Fig 11. Stabilized host-parasite-hyperparasite cycle with $K_M$ in 0.1 µM range.** Ribozyme concentrations of defined populations $R_1$, $R_2$, $R_3$ and $R_4$ [units on unspecified scale] are plotted against time $t$ [units on unspecified scale]. A habitat with highly efficient catalysts (ribozyme polymerase $R_1$, red, negative strand ribozyme $R_2$, light green) becomes fully stabilized in the presence of parasite-hyperparasite cycles, if the $K_M$ of the intermediate complexes $P\_F$ and $F\_F$ is decreased one order of magnitude in the 0.1 µM range (Model 3.2). Parasitism now is completely tamed ($P$ blue, $F$ dark green).

indistinguishable from the starting condition in Model 1. Since binding affinity under these conditions impacts directly on $K_D$ values and hence catalytic efficiency ($k_{cat}/K_M$) [35], incremental changes in the affinity of $F$ for itself and for $P$, by varying either the on-rate (fast association) or off-rate (slow dissociation) can impact significantly on the systems equilibrium [41,42]. Given what is currently known about template selection, modulating the on-rates appears to be the favoured option [43,44].

Several lines of experimental evidence suggest that self-binding within both host and parasite catalysts, with little cross-reactivity, is likely to happen. Firstly, just like proteins, RNAs exhibit a wide range of association rates, although even the fastest are significantly slower than those observed with proteins [42]. Nonetheless, RNA dimer formation is among the most rapid [42]. Secondly, the host and parasite-hyperparasite catalyst populations will diverge considerably, in both size and sequence, making the formation of homodimers or -multimers much more likely than that of heterodimers- or -multimers. Indeed, self-similarity increases the likelihood of physical interactions and spatially similar molecular structures interact preferentially [45,46]. In conclusion, our model is fully consistent with a growing sequence divergence of host and parasite catalysts, implying that the probability of self-binding increases, and the trend towards cross-reactivity diminishes.

Two temporally distinct scenarios can be envisaged; the parasite and hyperparasite immediately loose polymerase function because of the inherently high mutation load associated with the elevated replication rates of parasitic templates and acquire parasite binding specificity in a second step. Consistent with this notion, the same laws that govern substrate-enzyme interactions are also valid for protein-protein associations [41,47], and as an extension here for ribozyme polymerases [34,43,44,47]. This scenario is discussed in the main model (Fig 4). Alternatively, as outlined in the alternative model (Figs 4, S1 and S3) hyperparasite catalysts

with higher binding affinity for parasites are initially selected, but rapidly loose catalytic activity because of the accuracy-rate trade-off [35]. Subsequently, $F$'s catalytic activity is bound to degenerate, since $R_1$ mediated replenishment of the parasite pool is far more efficient than the one resulting from the poor catalytic activity of $F$.

In sum, these simulations show that parasite resistance can be modelled with biochemical precision, and it relies on hyperparasites acquiring parasite binding specificity, via at least two temporally distinct pathways.

## Discussion

We present a novel early-life model for autocatalytic ribozyme polymerase networks based on enzyme kinetics, more specifically the steady-state approximation. As a starting point, we defined a ribozyme polymerase with kinetic parameters drawn from in vitro models, which was capable of sustaining cyclic self-amplification and overcoming the error-threshold. We subsequently confirmed earlier models showing that these autocatalytic cycles are sustainable, as long as a sufficient nucleotide pool is available [23,31]. Inevitably, an efficient but promiscuous polymerase ribozyme with inherently low fidelity will generate molecular parasites that will overwhelm the habitat. However, because molecular parasitism is a chain process in principle, parasites of parasites (hyperparasites) will eventually appear. As long as these hyperparasites acquire parasite binding-specificity, an increasingly solid and molecularly specific parasite resistance will ensue. We demonstrate that this parasite resistance can be modelled with biochemical precision. Collectively, these findings add one more mechanism by which parasites can be tamed. It offers an attractive explanation for the universal and tight presence of molecular trios within the cellular hosts from prokaryotes to the virosphere and showcases the birth of parasite-resistance and a primitive molecular immunity (see below).

Once highly processive and efficient RNA catalysts are continuously provided to an experimental habitat, the host and parasite populations turn into an evolving ecosystem through Darwinian evolution, forming distinct host and parasite lineages that exhibit arms-race dynamics [37]. Resonating with this is the fact that self-replicating ribozyme networks consisting of three members display particularly high cooperative growth dynamics [7]. Intriguingly, there appears to be a trade-off between short and loosely folded sequences that are preferentially amplified (templates), and longer, tightly folded ones that are better ribozymes and are more resistant to hydrolytic decay [40]. This in-turn provides a rationale for splitting RNA communities into subgroups [7,37,40].

At the core of our current model is enzyme kinetics. The catalyst accuracy-rate trade-off analysis [35] applied to autocatalytic RNA networks [34] predicts the emergence of active ribozymes [10,11,46] with binding promiscuity and high catalytic activity, co-existing with subgenome-sized molecules characteristic of parasites with relatively lower catalytic activity [40]. Our model is fully consistent with these predictions. Furthermore, this enzyme kinetics-based model allows us to adapt these processes with high precision, since the input parameters can be varied at will. An incremental change in the affinity of the hyperparasite ribozyme for itself and for first generation parasite ribozymes can be achieved by varying association (fast association) and/or dissociation rates (slow dissociation) [41,42,47] although current models would favour the former option [43,44].

While primitive tripartite populations featuring rapidly degenerating catalysts likely represented a homeostatically stabilizing necessity, parasite communities within the population might have subsequently evolved more freely, with some becoming replicationally autonomous. Indeed, tripartite microbial populations composed of a host, a parasite and a hyperparasite are virtually ubiquitous [48], as evidenced for bacteria-bacteriophage-phage-satellites

[49,50], host-RNA viruses-defective interfering RNA viruses [51], and eukaryotic host-NCLDVs-virophages [52]. Furthermore, it is striking that the appearance of microbial immunity is associated with hyperparasites. Several recent studies have shown that phages and their satellites encode a set of diverse antiphage systems [53,54], and defective interfering RNAs activate innate immunity [55]. Phage satellite-encoded immune systems protect bacteria from phage predation and were proposed to be an integral component of innate immunity [56]. This notion could easily be extended to omnipresent, bona fide hyperparasites, such as endogenous retroviruses [57,58]. Therefore, there is likely a strong, ancestral link between hyperparasites and parasite immunity.

We previously developed a tripartite early-life host-parasite-hyperparasite framework based on Lotka Volterra (LV)-equations [31,59]. The present work arrives at similar results, showing that LV-based equations provide an adequate approximation for the more realistic, kinetics-based model developed here. Nonetheless, the present model has by design intrinsic limitations, notably the fact that we restrict the reaction to one intermediate polymerase-template complex [60]. Additionally, we assume that the catalytic efficiency of ribozyme polymerases will slowly approach that of proteinaceous holoenzymes and of mainstream, albeit complexed, multi-subunit enzymatic machineries.

In S1 Table, model assumptions, results and the underlying theoretical framework supporting the conclusions are compared for all main models and submodels. This helps untangling the deductions we have drawn from the built-in suppositions, and clearly separate the model conclusions from the underlying theoretical evidence.

Ribozymes polymerase-based replicators are subject to invasion by at least initially non-functional sequences that act as parasites. Molecular parasites will inevitably arise and persist, either by the generation of new random sequences, or as a result of replication errors of functional strands. The presence of molecular parasites will necessarily destroy the host habitat, unless a taming strategy is found. Two taming strategies have been widely studied, namely spatial self-organisation [18–24], and physical compartmentalisation inside protocells [25–28]. Our model bears several resemblances and differences to these two historical references. It is also based on a second-order catalysis [28], i.e. one molecular species, the positive strand ($R_1$), produces a negative stranded molecule ($R_2$) that serves as a template for amplification of the former ($R_1$). In line with these precedent models, the acquisition of hyperparasite-binding specificity for parasites in our system is likewise conferring the critical advantage to the host polymerase that permits habitat survival. While we make no implicit assumptions on spatial organisation and the existence of protocells, the tripartite habitat can be considered a special case of spatial self-organisation that prefigures protocells, because the three habitat constituents are individually interlinked, interdependent and essential for survival. Although molecular parasites have traditionally been viewed as a nuisance, our model argues for their integration into the host habitat rather their separation. Therefore, the tripartite habitat configuration conferring homeostatic stability is highly specific for our model and hence constitutes its quintessential novelty.

The Michaelis-Menten equation is a deterministic rate equation that assumes large numbers of enzyme and substrate molecules [61]. However, with low molecule numbers, time-course measurements display large stochastic fluctuations. Unsurprisingly, stochastic and deterministic approaches may predict different numerical values for the mean substrate, enzyme and complex concentrations over time, as well as different steady-state concentrations for a given set of rate constants. These differences are typically small for the Michaelis-Menten reaction, but significant with substrate inflow [61]. Therefore, stochastic models would not significantly modify our conclusions, except for situations where external substrate provision becomes dominant.

The qualitative results of the model would not change if the numerical constants and parameters were varied slightly. Indeed, the robustness of settings to variation was selected to

exhibit exactly that property. For example, all the parameters in Model 3 of Fig 10 can be slightly varied yet lead qualitatively to the same result. On the other hand, Fig 11 is one example where we lowered the $K_M$ from 1.0 to 0.1 μM to show that this would lead to a more efficient stabilization of the host-parasite-hyperparasite cycle and parasite taming. Nonetheless, qualitatively the results in Fig 11 are comparable to the ones in Fig 10. In future work we will investigate whether some parameters show trade-offs towards others, for instance the affinity of different RNA molecules versus their catalytic efficiency.

In principle, molecular parasitism is a chain process [62]. Indeed, the habitat of prototypical tripartite populations of host-parasites-hyperparasites as exemplified by bacteria, phages and phage satellites contains even smaller molecular parasites that could–at least in part–qualify as potential hyper-hyperparasites, namely a novel type of phage satellite, which is an autonomously replicating plasmid [63]. This entity does not integrate into the host's genome, does not carry phage genes and is smaller than any known phage satellite [63]. Yet, empirical evidence suggests that tripartite populations are the prevailing habitat type [62,64]. Moreover, mobile genetic elements are highly pleiotropic, exerting multiple, sometimes contrasting effects that defy their classification as pure or even dominant parasites [63–65].

## Methods or models

### 1 Michaelis-Menten and Briggs-Haldane kinetics

The enzyme kinetics generally known as Michaelis-Menten kinetics [29,30] is introduced as follows:

$$E + S \underset{k_{off}}{\overset{k_{on}}{\rightleftharpoons}} E\_S \overset{k_{cat}}{\rightharpoonup} E + P,$$

$E$ being the enzyme, $S$ the substrate, $E\_S$ the intermediate enzyme-substrate complex, and $P$ the product. If all the symbols here are meant to be concentrations in a given environment (habitat), then the following equations apply:

$$\frac{dS}{dt} = -k_{on}E \cdot S + k_{off}E\_S$$

$$\frac{dE}{dt} = -k_{on}E \cdot S + \left(k_{off} + k_{cat}\right)E\_S$$

$$\frac{dE\_S}{dt} = k_{on}E \cdot S - \left(k_{off} + k_{cat}\right)E\_S$$

$$\frac{dP}{dt} = k_{cat}E\_S \tag{1'}$$

An important assumption is that the concentration of $E\_S$ reaches the steady-state almost instantly (steady-state approximation). According to the Briggs-Haldane steady-state approximation [32], we therefore assume

$$\frac{dE\_S}{dt} = 0,$$

and thus:

$$k_{on}E \cdot S = (k_{off} + k_{cat})E\_S$$

Here $E$ is the free enzyme $E$, and $S$ the free substrate. If we write

$$E_{tot} = E + E\_S \tag{2}$$

then the equation above yields, with the Michaelis constant $K_M = \frac{k_{off}+k_{cat}}{k_{on}}$,

$$(E_{tot} - E\_S) \cdot S = K_M E\_S \tag{3}$$

and thus

$$E\_S = \frac{E_{tot} \cdot S}{K_M + S} \tag{4}$$

## 2 Michaelis-Menten kinetics in the RNA-world

Let us now consider a set of different RNA populations $\{R_1, R_2, \ldots, R_n\}$ in a primordial molecular RNA landscape, with the supply of nucleotides being provided by metabolism [23,31]. This metabolism is not further specified, and we assume that there is a limitation on the number of RNA molecules $\{R_1, R_2, \ldots, R_n\}$ that can be maintained (habitat restriction term, see below). These RNA molecules may be ribozyme polymerases if they exert a catalytic activity or merely templates.

**2.1 A simple three ribozyme system.** In this RNA world, a catalytic activity is exerted if a free ribozyme polymerase exhibiting Michaelis-Menten saturation kinetics [9] from population $R_1$ encounters another from $R_2$. The ribozyme $R_1$ uses the free molecule $R_2$ as template, and by forming a polymerase-template complex, catalyses the production of a third ribozyme population $R_3$ ($R_3$ being the negative strand of $R_2$). We now apply Michaelis-Menten kinetics, which entails that not only $R_1$ is a ribozyme polymerase, but also the substrate $R_2$, and even the product $R_3$. The Michaelis-Menten formalism is valid for this special example. If $R_1, R_2, R_3$ denote the concentrations of the total ribozyme masses, and $R_{1free}$, $R_{2free}$, $R_{3free}$ denote the concentrations of their respective free ribozyme masses, then $E = R_{1free}$, $S = R_{2free}$, and $P = R_3$. Then, Eq ($2'$) reads as follows: $R_{1free} = R_1 - R_1\_R_2$. But now, the free substrate $S = R_{2free}$ is also a function of the total substrate $R_2$ and its enzyme-substrate complex $R_1\_R_2$, i.e., $R_{2free} = R_2 - R_1\_R_2$. Eqs ($1'$) and ($3'$) from above now read as follows:

$$\frac{dR_3}{dt} = k_{cat}R_1\_R_2 \tag{1}$$

$$(R_1 - R_1\_R_2) \cdot (R_2 - R_1\_R_2) = K_M R_1\_R_2 \tag{3}$$

According to Briggs-Haldane it is assumed that the enzyme-substrate complex $R_1\_R_2$ reaches the steady-state very rapidly, $\frac{dR_1\_R_2}{dt} = 0$. Eq ($3''$) is a quadratic equation implying two solutions for unknown $R_1\_R_2$. Only one is physically relevant, the one guaranteeing positive concentrations of all free ribozymes, namely:

$$R_1\_R_2 = \frac{1}{2}(R_1 + R_2 + K_M) - \sqrt{\frac{1}{4}(R_1 + R_2 + K_M)^2 - R_1 R_2}$$

This solution of Eq ($3''$) is called the tight-binding or Morrison equation [66].

**2.2 The general case of $n$ ribozymes.** In the above special case of three ribozymes, we have assumed that only ribozymes $R_1$ and $R_2$ are forming an enzyme-substrate complex $R_1\_R_2$. Ribozyme $R_1$ does not bind to ribozyme $R_3$ or to itself $R_1$, nor does ribozyme $R_2$ bind to $R_1$ or to $R_3$. In the general case, where we have $n$ ribozymes $R_1, R_2, \ldots, R_n$, every ribozyme polymerase $R_i$ could potentially bind—through its role as enzyme—to a template (substrate) ribozyme $R_j$, forming an enzyme-template complex $R_i\_R_j$. In general, complex $R_i\_R_j$ is not equal to complex $R_j\_R_i$, since $R_i\_R_j$ is governed by the enzymatic activity of $R_i$, whereas $R_j\_R_i$ may be governed by the different enzymatic activity of $R_j$, which may not be the same.

Thus, in general, $n$ ribozymes can bind to $n^2$ different enzyme-template complexes $R_i\_R_j$, $i, j = 1, \ldots, n$.

$$R_{i_{free}} + R_{j_{free}} \underset{k_{off}^{i,j}}{\overset{k_{on}^{i,j} \qquad k_{cat}^{i,j}}{\rightleftharpoons}} R_i\_R_j \longrightarrow R_{j_{free}} + P_{i,j}$$

Eq (1′), describing the production rate of new products, now becomes

$$\frac{dP_{i,j}}{dt} = k_{cat}^{i,j} R_i\_R_j \tag{1}$$

In an $n$ ribozymes RNA world, the products $P_{i,j}$ can be found among the given ribozymes, i.e., $P_{i,j} \in \{R_1, R_2, \ldots, R_n\}$. Eq (2′), describing the relation between the total ($R_i$) and the free ($R_{i_{free}}$) concentration of ribozymes, now turns into the following Eq (2):

$$R_{i_{free}} = R_i - \left(\sum_{k=1}^{n} R_i\_R_k + R_k\_R_i\right) + R_i\_R_i \tag{2}$$

Again, we assume according to Briggs-Haldane that all ribozyme-template complexes $R_i\_R_j$ reach the steady-state very rapidly, $\frac{dR_i\_R_j}{dt} = 0$, $i, j = 1, \ldots, n$.

Given the Michaelis constants $K_M^{i,j} = \frac{k_{off}^{i,j} + k_{cat}^{i,j}}{k_{on}^{i,j}}$, Eq (3′) turns now, for every $i, j = 1, \ldots, n$, into

$$\left(R_i - \left(\sum_{k=1}^{n} R_i\_R_k + R_k\_R_i\right) + R_i\_R_i\right)\left(R_j - \left(\sum_{k=1}^{n} R_j\_R_k + R_k\_R_j\right) + R_j\_R_j\right) = K_M^{i,j} R_i\_R_j \tag{3}$$

Eq (3) is a nonlinear system of $n^2$ equations with $n^2$ unknown ribozyme-template complexes $R_i\_R_j$. Given the total concentrations, $R_1, R_2, \ldots, R_n$, this equation is solvable through an appropriate nonlinear multidimensional multistep algorithm. For this purpose, we propose a Newton iteration outlined as follows.

**2.3. Newton iteration for the solution of Eq (3).** Let $R\_R \in \mathbb{R}^{n^2}$ be a real valued $n^2$ dimensional vector, with

$$R\_R = [R_1\_R_1, R_1\_R_2, \ldots, R_1\_R_n, R_2\_R_1, R_2\_R_2, \ldots, R_2\_R_n, R_3\_R_1 \ldots, R_n\_R_n]^T$$

Let $F : \mathbb{R}^{n^2} \longrightarrow \mathbb{R}^{n^2}$, $R\_R \longmapsto F(R\_R)$, be a $n^2$-multidimensional function
$F(R\_R) = [F_1(R\_R), F_2(R\_R), \ldots, F_l(R\_R), \ldots, F_{n^2}(R\_R)]^T$ with

$$F_l(R_R) = K_M^{i,j} R_i\_R_j - \left(R_i - \left(\sum_{k=1}^{n} R_i\_R_k + R_k\_R_i\right) + R_i\_R_i\right)\left(R_j - \left(\sum_{k=1}^{n} R_j\_R_k + R_k\_R_j\right) + R_j\_R_j\right),$$

$l = 1,\ldots, n^2$, where the indices $i, j$ themselves are functions of index $l$. If we define these functions in the following way:

$$j(l) = \begin{cases} (l \bmod n), & if \ (l \bmod n) > 0 \\ (l \bmod n) + n, & if \ (l \bmod n) = 0 \end{cases}$$

$$i(l) = \frac{(l - j(l))}{n} + 1,$$

then the $l$th component of vector $R\_R$, is well-defined as $(R\_R)_l = R_{i(l)}\_R_{j(l)}$, and also the $l$th component of $F$, $F_l(R\_R)$ is well defined for every index $l = 1,\ldots,n^2$. Now we state that a vector $R\_R = [R_1\_R_1, R_1\_R_2, \ldots., R_n\_R_n]^T$ is a solution of (3), if an only if $F(R\_R) = 0$, i.e. $F_l(R\_R) = 0$ for every $l = 1,\ldots,n^2$. The classic multidimensional Newton iteration algorithm needs an explicit expression for of the Jacobian $J$ of $F(R\_R)$:

$$J_{l,m} = \frac{\partial F_l(R\_R)}{\partial (R\_R)_m}, l, m = 1 \ldots n^2$$

The full derivation of the formula for the Jacobian is straight forward, therefore we just give here the final result. Let $\delta(i,j) = \begin{cases} 1, if \ i = j \\ 0, \quad else \end{cases}$ be the Kronecker symbol, then the explicit formula for the Jacobian is given as follows. For $l, m = 1,\ldots,n^2$:

$$J_{l,m} = \frac{\partial F_l(R\_R)}{\partial (R\_R)_m} = K_M^{i(l),j(l)}\delta(l, m) +$$
$$+ R_{j_{free}}(\delta(i(m), i(l)) + \delta(j(m), i(l)) - \delta(i(m), i(l))\delta(j(m), i(l))) +$$
$$+ R_{i_{free}}(\delta(i(m), j(l)) + \delta(j(m), j(l)) - \delta(i(m), j(l))\delta(j(m), j(l)))$$

In this formula, the free ribozymes, $R_{i_{free}}$, are calculated according to (2).

Now we are able to write down the full Newton iteration algorithm. Given some initial set of total ribozyme concentrations $\{R_1, R_2,\ldots,R_n\}$, we start assuming that no ribozyme-template complex has been built yet; i.e. $R_{i_{free}} = R_i, i = 1, .., n$, and $(R\_R)_l = 0, l = 1,\ldots, n^2$. The initial vector of ribozyme-template complexes therefore equals to zero: $(R\_R)^{(0)} = 0$. After $k$ iterations, the resulting vector of ribozyme-template complexes is then calculated by:

$$(R\_R)^{(k+1)} = (R\_R)^{(k)} - (J)^{-1}F((R\_R)^{(k)}), k = 0, 1, 2, \ldots$$

Here $(J)^{-1}$ is the is $n^2 \times n^2$ inverse matrix of the Jacobian $J$. Normally, this sequence of iterations convergences to a solution $(R\_R)^{(k)} \to (R\_R)^* = R\_R$, with $F((R\_R)^*) = 0$. The entries of the resulting vector $R\_R$ are then concentrations of all of the possible ribozyme-template complexes $(R\_R)_l = R_{i(l)}\_R_{j(l)}$, $l = 1,\ldots, n^2$, solving Eq (3).

## 3. Template based autocatalytic cycles

**3.1 Preamble.** Reconsider the set of populations of different RNA molecules $\{R_1, R_2,\ldots, R_n\}$ introduced above. The ribozymes form a catalytic cycle, if

$$R_1 \to^c R_2 \to^c R_3 \to^c \ldots \to^c R_n \to^c R_1,$$

where the symbol "$A \to^c C$" means: ribozyme $A$ catalyzes ribozyme $C$. The catalytic production of $C$ is called *template based* if an encounter with a template ribozyme $B$ is needed for this

process, i.e. *A* needs to form, together with *B*, an enzyme-template intermediate complex *A_B* in order to produce *C*. We write "$A\_B \to^c C$". If the autocatalytic cycle reproduces not only all ribozymes carrying an enzymatic role, but also all ribozymes serving as templates along its way then we call it a *template based autocatalytic cycle*. The following example meets all the criteria of such a template based autocatalytic cycle:

$$R_1 \Rightarrow (R_1\_R_1) \to^c R_2 \Rightarrow (R_2\_R_1) \to^c R_3 \Rightarrow (R_2\_R_3) \to^c R_4 \Rightarrow (R_4\_R_1) \to^c R_1$$

This reads as follows, a ribozyme $R_1$ forms an enzyme-template complex with itself, $R_1\_R_1$, producing a new copy of ribozyme $R_2$. $R_2$ too forms an enzyme-template complex with $R_1$, $R_2\_R_1$, producing ribozyme $R_3$. $R_2$ now forms another enzyme-template complex with $R_3$, $R_2\_R_3$, producing $R_4$, which in turn produces again $R_1$ by forming a catalytic enzyme-template complex $R_4\_R_1$ with $R_1$. Now we are able to model primitive template based autocatalytic cycles in an RNA landscape, with a given, but not further specified metabolism providing a limited supply of nucleotides.

**3.2. Model 1: the host cycle.** Let ribozyme $R_1$ be a population of positive strand ribozyme polymerases, and $R_2$ be the population of its negative strand, then we can model the following template based autocatalytic cycle:

Model 1

$$R_1 \Rightarrow (R_1\_R_1) \to^c R_2 \Rightarrow (R_1\_R_2) \to^c R_1 \tag{1.0}$$

The replication dynamics is governed by the following ODEs:

$$\frac{dR_1}{dt} = k_{cat}^{1,2} R_1\_R_2 \Pi - d_{R_1} R_1 \tag{1.1}$$

$$\frac{dR_2}{dt} = k_{cat}^{1,1} R_1\_R_1 \Pi - d_{R_2} R_2, \tag{1.2}$$

Here $d_{R_i}$ are the decay rates of populations $R_i$, $i = 1,2$, and $\Pi = \left(1 - \frac{R_1 + R_2}{K}\right)$ is the habitat restriction term. $K$ is the maximum number of $R_1$ or $R_2$ RNA molecules the habitat is able to maintain through the given metabolism. Given the total concentrations $R_1$ and $R_2$ we assume using Briggs-Haldane that the enzyme-template intermediate complexes $R_1\_R_1$ and $R_1\_R_2$ are reaching their steady-state at every point in time very rapidly. In other words, we assume that the process of attaining the steady-state for the enzyme-template intermediate complexes is by some orders of magnitudes faster than the production of new ribozymes. In order to calculate all of the possible enzyme-template complexes $R_i\_R_j$ through the Newton iteration algorithm, as outlined above, we need to attribute values to $k_{on}^{i,j}, k_{off}^{i,j}, k_{cat}^{i,j}$, along the Michaelis constants $K_M^{i,j} = \frac{k_{off}^{i,j} + k_{cat}^{i,j}}{k_{on}^{i,j}}, i,j = 1.2$. E.g.

$$k_{on} = \begin{bmatrix} 0.2 & 0.19 \\ \varepsilon & \varepsilon \end{bmatrix}, k_{off} = \begin{bmatrix} \eta & \eta \\ \eta & \eta \end{bmatrix}, k_{cat} = \begin{bmatrix} 0.1 & 0.08 \\ 0 & 0 \end{bmatrix}, K_M = \begin{bmatrix} 1.0 & 0.947 \\ \eta/\varepsilon & \eta/\varepsilon \end{bmatrix}$$

We further note that in this model ribozyme $R_2$ does not form an enzyme-template complex with $R_1$, nor with itself, assuming that $R_2$−the negative strand of $R_1$−does not have catalytic activity. Since the Newton algorithm needs Michaelis constants for all possible enzyme-template complexes $R_i\_R_j$, we assume here that $k_{on}^{2,1} = k_{on}^{2,2} = \varepsilon$ is a very low number greater than zero, and also $k_{off}^{2,1} = k_{off}^{2,2} = \eta > 0$. We choose $\varepsilon$ and $\eta$ so that formation of enzyme-template complexes $R_2\_R_1, R_2\_R_2$ is possible, but always at very low concentration and also dissociating very rapidly; e.g., $\varepsilon = 10^{-5}, \eta = 10^{-1}$. By choosing appropriate values for all other

parameters (e.g. $d_{R_i} = d_{R_2} = 0.005$, $K = 20.0$, $R_{1\,initial} = 1.0$, $R_{2\,initial} = 1.0$). The ODEs (1.1), (1.2) of Model 1 are solved numerically by simple timestep integration. Note that for every timestep $t_k = t_0 + k\Delta t$ a calculation of all the possible enzyme-template complexes $R_i\_R_j$ through the Newton iteration algorithm is needed.

**3.3. Model 2: the emergence of parasitism.** Autocatalytic host cycles (1.0) inevitably are subject to molecular parasitism [16,17]. In (1.0), the ribozyme polymerase $R_1$ is replicating $R_2$ and itself. Such a replicase cannot be so specific that it only replicates $R_1$ and $R_2$, since under such specific conditions $R_1$ would quickly lose its catalytic activity due to mutations it is subject to [16,17]. Therefore, ribozyme polymerase $R_2$ will inevitably be parasitized by RNA templates, typically of subgenomic size [37,38], say $R_3 = P$ (parasites):

$$P = R_3 \Rightarrow (R_2\_R_3) \to^c R_3 = P \tag{2.0}$$

This motivates following additional equation

$$\frac{dR_3}{dt} = k_{cat}^{1,3} R_2\_R_3 \Pi - d_{R_3} R_3 \tag{2.1}$$

The habitat restriction term $\Pi$ extends to

$$\Pi = \left(1 - \frac{R_1 + R_2 + {}^{R_3}/_{q_{R_3}}}{K}\right),$$

where $q_{R_3} > 1$, since parasites $R_3 = P$ are more efficient at using the habitat resources, because of their smaller size that confers a replicative advantage [38]. A ribozyme polymerase $R_1$ that engages in replicating $R_3 = P$, building an enzyme-template complex $R_1\_R_3$, cannot simultaneously form enzyme-template complexes with $R_1\_R_2$ or $R_1\_R_2$. Thus, $R_3$ parasitizes the whole autocatalytic cycle (1.0). Here, for simplicity, we do not distinguish between positive and negative strand parasites.

We model two parasite sources, the most probable being the imperfect fidelity of polymerase $R_1$. Rather than producing a viable RNA molecule $R_2$ or $R_1$, it may, at a specific mutation rate, $\mu > 0$, produce a non-viable mutant having more similarity to a parasite $P = R_3$, rather than to a viable RNA molecule of the host cycle (1.0). Parasite invasion from a neighbouring habitat can be viewed as an alternative. The parameter $\alpha \geq 0$ denotes the parasite inflow per unit of time.

The ODE equations for Model 2 are as follows:

$$\frac{dR_1}{dt} = (1 - \mu) k_{cat}^{1,2} R_1\_R_2 \Pi - d_{R_1} R_1 \tag{2.2}$$

$$\frac{dR_2}{dt} = (1 - \mu) k_{cat}^{1,1} R_1\_R_1 \Pi - d_{R_2} R_2, \tag{2.3}$$

$$\frac{dR_3}{dt} = \left(k_{cat}^{1,3} R_1\_R_3 + \mu\left(k_{cat}^{1,2} R_1\_R_2 + k_{cat}^{1,1} R_1\_R_1\right)\right)\Pi + \alpha - d_{R_3} R_3 \tag{2.4}$$

Here we assign values corresponding to the ones used in Model 1: $d_{R_1} = d_{R_2} = d_{R_3} = 0.005$, $K = 20.0$, $q_{R_3} = 5.0$, $R_{1\,initial} = 1.0$, $R_{2\,initial} = 1.0$, $R_{3\,initial} = 0.0$, $\varepsilon = 10^{-5}$, $\eta = 10^{-1}$, and

$$k_{on} = \begin{bmatrix} 0.2 & 0.19 & 0.2 \\ \varepsilon & \varepsilon & \varepsilon \\ \varepsilon & \varepsilon & \varepsilon \end{bmatrix}, k_{off} = \begin{bmatrix} \eta & \eta & \eta \\ \eta & \eta & \eta \\ \eta & \eta & \eta \end{bmatrix}, k_{cat} = \begin{bmatrix} 0.1 & 0.08 & 0.1 \\ 0 & 0 & 0 \\ 0 & 0 & 0 \end{bmatrix}, K_M = \begin{bmatrix} 1.0 & 0.947 & 1.0 \\ \eta/\varepsilon & \eta/\varepsilon & \eta/\varepsilon \\ \eta/\varepsilon & \eta/\varepsilon & \eta/\varepsilon \end{bmatrix}$$

**3.4. Model 3: the emergence of hyperparasitism.**   Parasitism is a chain process, molecular parasites of even smaller size than the first-generation parasites are bound to emerge over time [37,38]. The model stipulates that the ribozyme polymerase $R_1$ replicates host templates $R_1$ faithfully, the enzyme-template complexes $R_1\_R_1$ and $R_1\_R_2$ only catalyze $R_2$ and $R_1$, respectively. If replicase $R_1$ binds to a parasite template $R_1\_R_3$, it not only produces another copy of $R_3$, or a negative strand of $R_3$, respectively, which we again for simplicity do not distinguish here, and also another, smaller RNA molecule $R_4 = F$, which we call here the hyperparasite, $F$ being smaller than $R_3$, but partially overlapping with it.

The hyperparasite cycle is triggered from the following process

$$R_1 \Rightarrow (R_1\_P) \rightarrow^c F(1 \,\&\, 2 \rightarrow 3)$$

and has the following form:

$$F \Rightarrow (P\_F) \rightarrow^c F$$

$$F \Rightarrow (F\_F) \rightarrow^c F \tag{3.0}$$

The hyperparasite $F$ binds to an enzyme-template complex $P\_F$ or to itself $F\_F$, which implies that $F$ is capable of replicating itself. Again, for simplicity, we do not distinguish between positive and negative strand hyperparasites $R_4 = F$, assuming both are present. But since the production of $F$ is already guaranteed through the trigger process (1&2→3), such a process may degenerate to produce futile intermediate complexes [67] $P\_F$ or $F\_F$:

$$F \Rightarrow (P\_F) \rightarrow^c nil$$

$$F \Rightarrow (F\_F) \rightarrow^c nil \tag{3.0}$$

$F = R_4$ is a hyperparasite, because it parasitizes the parasite $P$ while bound to a complex $P\_F$. A parasite molecule $P$ bound to $F$ is not able at the same time to bind to the ribozyme polymerase $R_1$, and therefore cannot be replicated anymore. In order to be effective for the host, the hyperparasite requires to have binding specificity for parasite molecules, $R_3 = P$: $F$ should form intermediate enzyme-template complexes only with $R_3 = P$ molecules or with itself.

Now we are able to write down the ODE equations for Model 3, the host-parasite-hyperparasite model.

Model 3

$$\frac{dR_1}{dt} = (1-\mu)k_{cat}^{1,2}R_1\_R_2\Pi - d_{R_1}R_1 \tag{3.1}$$

$$\frac{dR_2}{dt} = (1-\mu)k_{cat}^{1,1}R_1\_R_1\Pi - d_{R_2}R_2, \tag{3.2}$$

$$\frac{dR_3}{dt} = \left(k_{cat}^{1,3}R_1\_R_3 + \mu\left(k_{cat}^{1,2}R_1\_R_2 + k_{cat}^{1,1}R_1\_R_1\right)\right)\Pi + \alpha - d_{R_3}R_3 \tag{3.3}$$

$$\frac{dR_4}{dt} = \beta k_{cat}^{1,3} R_1 R_3 \Pi - d_{R_4} R_4 \tag{3.4}$$

The habitat restriction parameter extends here to

$$\Pi = \left(1 - \frac{R_1 + R_2 + {}^{R_3}/_{q_{R_3}} + {}^{R_4}/_{q_{R_4}}}{K}\right),$$

Here we again assign values accordingly to the ones in Model 1 and 2:
$d_{R_1} = d_{R_2} = d_{R_3} = d_{R_4} = 0.005$, with
$q_{R_3} = 5.0$, $q_{R_4} = 10.0$, $K = 20.0$, $R_{1\,initial} = 1.0$, $R_{2\,initial} = 1.0$, $R_{3\,initial} = 0.0$, $R_{4\,initial} = 0.0$,
mutation rate $\mu = 0.01$, parasite inflow per unit of time, $\alpha = 0.001$, $\beta = 1.5$ ($\beta$ is here a parameter
quantizing the trigger process (1&2→3)), $\varepsilon = 10^{-5}$, $\eta = 10^{-1}$, and

$$k_{on} = \begin{bmatrix} 0.2 & 0.19 & 0.2 & \varepsilon \\ \varepsilon & \varepsilon & \varepsilon & \varepsilon \\ \varepsilon & \varepsilon & \varepsilon & 0.1 \\ \varepsilon & \varepsilon & \varepsilon & 0.1 \end{bmatrix}, k_{off} = \begin{bmatrix} \eta & \eta & \eta & \eta \\ \eta & \eta & \eta & \eta \\ \eta & \eta & \eta & \eta \\ \eta & \eta & \eta & \eta \end{bmatrix}, \begin{bmatrix} 0.1 & 0.08 & 0.1 & 0 \\ 0 & 0 & 0 & 0 \\ 0 & 0 & 0 & 0 \\ 0 & 0 & 0 & 0 \end{bmatrix},$$

$$K_M = \begin{bmatrix} 1.0 & 0.947 & 1.0 & \eta/\varepsilon \\ \eta/\varepsilon & \eta/\varepsilon & \eta/\varepsilon & \eta/\varepsilon \\ \eta/\varepsilon & \eta/\varepsilon & \eta/\varepsilon & 1.0 \\ \eta/\varepsilon & \eta/\varepsilon & \eta/\varepsilon & 1.0 \end{bmatrix}$$

The values for Model 3 are exactly the same as in Model 2, except for the parameters affecting the hyperparasite $R_4 = F$.

**3.5 Model 3.1, μM-range Michaelis constants for P_F and F_F complexes.** Here we chose the following parameters for the binding of the hyperparasite $R_4 = F$ with $R_3 = P$ or with itself: $k_{on}^{3,4} = k_{on}^{4,4} = 0.1$, $k_{off}^{3,4} = k_{off}^{4,4} = 0.1$, and hence the Michaelis constants $K_M^{3,4} = K_M^{4,4} = 1.0$.

**3.6 Model 3.2, sub-μM-range Michaelis constants for P_F and F_F complexes.** Now if the binding of the complexes P_F, F_F is further strengthened, by e.g. increasing $k_{on}^{3,4} = k_{on}^{4,4} = 1.0$, and so lowering the Michelis constants $K_M^{3,4} = K_M^{4,4} = 0.1$, the situation for the host cycle can even be further stabilized, with a lower population of parasites $P$ and less molecules bound to the P_F and F_F complexes. Thus, if the hyperparasite $F$ does not or only very weakly interact with the larger host ribozyme polymerases $R_1$ and $R_2$, but does bind with higher affinity to parasite molecules $P$ and also to itself, then it is offering solid stability to the host cycle in the presence of parasites $P$.

## Supporting information

**S1 Definition of terms and notations. Key terms and model parameters are defined, listed in alphabetical order and references are indicated where appropriate.** The model parameters (1st column), the values (2nd column) and a short textual explanation (3rd column) are listed for the main models and for the submodels. Reference to *Methods or Models*, and *S1 Supporting Information* is made where appropriate.
(DOCX)

**S1 Fig. The alternative model 3.0.1 with reduced catalytic activity and $K_M$ values in μM range.** Ribozyme concentrations of defined populations $R_1$, $R_2$, $R_3$ and $R_4$ [units on unspecified scale] are plotted against time $t$ [units on unspecified scale]. The catalytic activity is reduced ($k_{cat}$ 0.04 sec$^{-1}$). The ribozyme polymerase $R_1$ (red), the negative strand ribozyme $R_2$ (light green), the parasite $P$ (blue), and the hyperparasite $F$ (dark green).
(TIF)

**S2 Fig. The null catalytic main model 3.1 with $K_M$ values in μM range.** Ribozyme concentrations of defined populations $R_1$, $R_2$, $R_3$ and $R_4$ [units on unspecified scale] are plotted against time $t$ [units on unspecified scale]. The catalytic activity of the main model is null ($k_{cat}$ 0.0 sec-1). The ribozyme polymerase $R_1$ (red), the negative strand ribozyme $R_2$ (light green), the parasite $P$ (blue), and the hyperparasite $F$ (dark green).
(TIF)

**S3 Fig. The alternative model 3.0.2 with reduced catalytic activity and $K_M$ values in 0.1 μM range.** Ribozyme concentrations of defined populations $R_1$, $R_2$, $R_3$ and $R_4$ [units on unspecified scale] are plotted against time $t$ [units on unspecified scale]. The catalytic activity is reduced ($k_{cat}$ 0.04 sec-1), and $K_M$ values are 0.14 μM. The ribozyme polymerase $R_1$ (red), the negative strand ribozyme $R_2$ (light green), the parasite $P$ (blue), and the hyperparasite $F$ (dark green).
(TIF)

**S4 Fig. The null catalytic main model 3.2 and $K_M$ values in 0.1 μM range.** Ribozyme concentrations of defined populations $R_1$, $R_2$, $R_3$ and $R_4$ [units on unspecified scale] are plotted against time $t$ [units on unspecified scale]. The catalytic activity of the main model is null ($k_{cat}$ 0.0 sec-1), and $K_M$ values are 0.14 μM. The ribozyme polymerase $R_1$ (red), the negative strand ribozyme $R_2$ (light green), the parasite $P$ (blue), and the hyperparasite $F$ (dark green).
(TIF)

**S1 Table.** Model assumptions (3rd column), model results (4th column) and the underlying theoretical framework (5th column) are shown for individual models (1st column) and sub-models (2nd column).
(XLSX)

**S1 Supporting Information. It is the composite label for supplementary data that comprise three data sets, i) S1 Definition of terms and notations, ii) the main file S1 Supporting Information describing the alternative model with reduced catalytic activity featuring the 4 supporting S1–S4 Figs, as well as the References, and iii) S1 Table.**
(ODT)

## Acknowledgments

We thank Thomas Curran for discussions.

## Author Contributions

**Conceptualization:** Magnus Pirovino, Bernard Conrad.

**Formal analysis:** Magnus Pirovino, Bernard Conrad.

**Investigation:** Magnus Pirovino, Joseph A. Curran, Bernard Conrad.

**Methodology:** Magnus Pirovino, Christian Iseli, Bernard Conrad.

**Software:** Christian Iseli.

**Supervision:** Magnus Pirovino, Bernard Conrad.

**Validation:** Magnus Pirovino, Christian Iseli, Joseph A. Curran, Bernard Conrad.

**Visualization:** Christian Iseli, Bernard Conrad.

**Writing – original draft:** Magnus Pirovino, Joseph A. Curran, Bernard Conrad.

**Writing – review & editing:** Magnus Pirovino, Joseph A. Curran, Bernard Conrad.

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
