## [Decision Letter · Decision Letter 0]

8 Aug 2024

Dear Dr Conrad,

Thank you very much for submitting your manuscript "Biomathematical enzyme kinetics model of prebiotic autocatalytic RNA networks: degenerating parasite-specific hyperparasite catalysts confer parasite resistance and herald the birth of molecular immunity" for consideration at PLOS Computational Biology.

As with all papers reviewed by the journal, your manuscript was reviewed by members of the editorial board and by several independent reviewers. In light of the reviews (below this email), we would like to invite the resubmission of a significantly-revised version that takes into account the reviewers' comments.

We cannot make any decision about publication until we have seen the revised manuscript and your response to the reviewers' comments. Your revised manuscript is also likely to be sent to reviewers for further evaluation.

Sincerely,

Jacob G. Scott, MD

Academic Editor

PLOS Computational Biology

Tobias Bollenbach

Section Editor

PLOS Computational Biology

Reviewer's Responses to Questions

**Comments to the Authors:**

Reviewer #1: Overall, this article is well-written, containing interesting and novel science. Qualitatively, it explores how molecular parasites seem to be an inevitable issue for autocatalytic polymers at the origins of life, but also how molecular hyperparasites seem to solve this issue, saving the original polymers from dying out due to the initial parasites. Quantitatively, the authors express these points through simulated enzyme kinetic equations of Michaelis-Menten type, employing the perpetual steady-state approximation for the concentrations of intermediate complexes. This article exemplifies the goals of this journal, combining qualitative biological insight with mathematical rigor; in particular the math is explained with great clarity. Therefore, I would recommend it for publication, but I believe the article would benefit from some minor revisions. Specifically, a few brief updates - which I will note in detail below - would enhance the accessibility of this article to the broader community of scientists across the spectrum from biological to physical and mathematical. This article is already of high quality, and some of the following points might be addressed in included references, but the inclusion of one or two sentences here and there would make the article more self-contained:

1. The authors use biological terminology with specific meanings in this context. It would be helpful to briefly define these terms in the text, especially if the terms are central to the article, and if they could have other scientific or everyday meanings (e.g. molecular parasite, promiscuous, negative strand). This point, in particular, will assist scientists from a more physical or mathematical background understand the important qualitative results of the article.

2. The authors mention that they use relevant experimental values for numerical constants when available. If possible, it would be helpful to mention how the qualitative results of the article would change - if at all - when the numerical constants are varied.

3. The authors mention in the Discussion that molecular parasitism is a “chain process”, which seems to imply the existence of hyper-hyper-parasites and so on. Does the process continue to arbitrary nth-degree parasites? If so, does that affect the qualitative results of this article? If not, why does the process stop? Because the typical size of parasites gets smaller down the chain? It would be helpful to briefly clarify this question, as it seems to be a central concept to the article.

4. The authors ordinary differential equations, encoding a deterministic time evolution of concentrations of molecular species. It could be useful for the authors to briefly address if stochastic effects would potentially alter the conclusions of their study.

Reviewer #2: attached

Reviewer #3: The authors study prebiotic RNA amplification using the framework of Michaelis-Menten/Briggs-Haldane kinetics (i.e., they apply a quasi-steady-state assumption for the ODE describing the time evolution of the enzyme substrate complex). Based on the model the authors propose mechanisms how a host-parasite-hyperparasite trio can confer parasite resistance. The focus of this work is clearly not computational. The author use no methods which go beyond standard solvers for ODEs (timestep integration) and nonlinear algebraic equations (Newton iteration).

(A) The main text is very difficult to follow.

- The Methods Section is located at the very end of the manuscript. Unfortunately, it is impossible to understand the results without having read the Methods-Section before.

- Relevant notations have to be rigorously introduced before being used, e.g., kcat, kon, koff, KM, appear first on p. 4 but are defined on p. 9

- The notation of catalyzed reactions has to be rigorously introduced (e.g., 1 ⇒(1_1) → 2 ⇒ (1_2) → 1 (1.0)). It is first used on p. 5 but formally introduced on p. 13

- I suggest provide a Figure in the main text which summarizes all considered models.

- Relevant termini have to be clearly defined, e.g., what is meant by “promiscuous substrate binding”

- Please define “degenerate”.

- Please define “parasite resistance”.

- Inconsistent cross references further complicate the understanding of the text: For an overview of the used models the authors refer to Supplementary Figure 2. This Figure visualizes 3 models, referred to as “Model 1”, “Model 2”, “Model 3”. However, in the results section the authors refer to “Model 2.1”, “Model 2.2”, “Model 3”, “Model 3.1”, “Model 3.2”, “main model” etc.

(B) The descriptions of key results lack relevant details

- “The catalyst accuracy-rate trade-off

analysis [35] applied to autocatalytic RNA networks [34] predicts the emergence of active ribozymes

[10,11,46] with binding promiscuity and high catalytic activity, co-existing with subgenome-sized

molecules characteristic of parasites with relatively lower catalytic activity [40]. This is precisely what

our model predicts.” Please explain why this is a model prediction. I have the feeling that it is rather an assumption put into the model (reflected in the parameter choices).

- “In conclusion, with growing sequence divergence of host and parasite catalysts the probability of self-binding increases, and the trend towards cross

reactivity diminishes, precisely as suggested by our model.” Please describe in detail how this conclusion can be deduced from the model and under which assumptions. As far as I understand the model does not explicitly account for sequence similarity and its quantitative impact on physical interactions. Therefore, I am puzzled how the authors obtain this specific mechanistic conclusion from a rather phenomenological modeling approach.

-“In sum, these simulations show that parasite resistance can be modelled with biochemical

precision, and it relies on hyperparasites acquiring parasite binding specificity, via at least two

temporally distinct pathways.” This conclusion sounds odd. The authors do not compare results to data, therefore it remains unclear to me what they mean by “biochemical precision”.

- A main argument of the proposed reasoning is that the hyperparasite F has a very high specifity for the parasite P and that P_F complexes from with a higher probability compared to F_R1 complexes. Please elaborate on the question why such a constellation is more probable to arise during evolution compared to a situation where the parasite is highly specific for the host and P_R1 complexes from with a higher probability compared to P_F complexes. I agree with the conclusion “As long as these

hyperparasites acquire parasite binding-specificity, an increasingly solid and molecularly specific

parasite resistance will ensue” However, I do not see why this scenario is probable to arise during molecular evolution.

- All simulation results are dimensionless which makes it very difficult to interpret them and to compare the different considered scenarios.

-“the abundance of these two molecular species increases rapidly”. What does “rapidly” mean? The simulation provides no time unit.

- The caption of each figure should provide information about the used parameters

- “Figure 1 shows a graphical representation of this model assuming realistic starting kinetics and habitat restriction parameters (described in Methods or Models).” Please elaborate on what “realistic” means or refer the reader to a specific subsection of the Methods part where this issue is discussed.

- Fig 2A: How do the dynamics look like on a longer time scale. Does the system approach an equilibrium state?

- “As a corollary, F will be a highly efficient template, yet a poor or non-functional ribozyme capable

of self-amplification.” Please elaborate on this statement. Does the same reasoning also apply to P?

- “Secondly, in order for hyperparasitism to be highly efficient, it is required that F has a high specificity for P; i.e. in the presence of both R1 and P it binds mainly or exclusively to the latter” Please justify this statement in more detail and provide a formal proof in terms of the proposed models.

(C) More detailed explanations could significantly improve the readability of the text.

- “However, this process is not subject to evolutionary pressure and is bound to degenerate ( ⇒ (_) → , ⇒ (_) → (3.0)).” Please explain this statement in more detail.

- l. 189 what is meant by “and the level of P remains important.”

- Supplement: Bottom of page 10: Why is the solution R1_R2=1/2(R1+R2+KM)+sqrt(…) not considered?

- I do not see how the procedure described in section 3.3 differs from what is realized by standard solvers provided by any numerical software.

- Please justify the algebraic expression of the habitat restriction constant or provide a suitable reference.

- The parameter choices presented in the supplement have to be justified / motivated in more detail.

(D) I miss a section which systematically relates the model output to data or to experimental observations.

(E) Parasitism and hyperparasitism have been studied for a long time. I miss a systematic comparison to previous works and a paragraph summarizing the key novelties of this work.

(F) The abstract has to be reworked and clearly distinguish between model assumptions and drawn conclusions. Colloquial terms should be replaced by rigorous scientific language.

Minor

-“A complete autocatalytic replication cycle comprises the initial production of R2 by two R1 ribozyme molecules, and the subsequent reproduction of R1 by R1 using template R2” Please provide a reference supporting this mechanism

-“For a number of reasons they

148 are inseparably associated with all life forms [16,17], with the most important being the accuracy-rate

149 trade-off [34,35].” Please explain this statement in more detail, e.g., in the introduction section

-“In general, enzymes operate orders of magnitude (kcat/KM of ∼105 M–1s–1) below the

150 diffusion rate (108-109 M-1s-1). This implies promiscuous substrate binding, high catalytic activity, and as

151 a secondary trade off, a high mutation rate generating a large variance in the quasispecies: i.e. speed

152 being more important than enzyme accuracy [36].” Please explain this statement in more detail, e.g., in the introduction section

-“Indeed, this is precisely what

178 the accuracy-rate trade-off predicts, with selectivity implying the ability to discriminate between two

179 templates when both are present [35], whereas rate denotes speed [35,36].” This senetenc eis hard to digest.

-“Ground state

180 discrimination means that specificity is achieved mainly through substrate binding, which imposes

181 strong accuracy/rate trade-offs [34,35].” Please explain this statement in more detail, e.g., in the introduction section

- “Improvements in selectivity mediated by tighter cognate

182 substrate binding invariably leads to lower catalytic efficiency (parallel decreases in the constants for

the cognate substrates KM

cog and kcat

183 cog [34,35]).” Please explain this statement in more detail, e.g., in the introduction section

- Fig. 3A “This already happens if intermediate complexes P_F and F_F have a KM in the �M range” To facilitate the interpretation please state the KM values of all catalytic processes involved in Model 3. The same applies to Fig 3B.

- “Given what is currently known

197 about template selection, modulating the on-rates appears to be the favoured option [43,44].” Please shortly explain why

- “We subsequently confirmed earlier models

228 showing that these autocatalytic cycles are sustainable, as long as a sufficient nucleotide pool is

229 available.” Please provide references to the “earlier models”.

**Have the authors made all data and (if applicable) computational code underlying the findings in their manuscript fully available?**

Reviewer #1: Yes

Reviewer #2: Yes

Reviewer #3: Yes

PLOS authors have the option to publish the peer review history of their article (what does this mean?). If published, this will include your full peer review and any attached files.

Reviewer #1: No

Reviewer #2: No

Reviewer #3: No
---

## [Decision Letter · Decision Letter 1]

3 Dec 2024

Dear Dr Conrad,

We are pleased to inform you that your manuscript 'Biomathematical enzyme kinetics model of prebiotic autocatalytic RNA networks: degenerating parasite-specific hyperparasite catalysts confer parasite resistance and herald the birth of molecular immunity' has been provisionally accepted for publication in PLOS Computational Biology.

Best regards,

Jacob Scott, MD

Academic Editor

PLOS Computational Biology

Tobias Bollenbach

Section Editor

PLOS Computational Biology

Feilim Mac Gabhann

Editor-in-Chief

PLOS Computational Biology

Jason Papin

Editor-in-Chief

PLOS Computational Biology

The authors have done a great job addressing the requested reviews, and the reviewers are now largely satisfied. One reviewer does note a few small edits however that I do feel would strengthen the manuscript. I do not believe we need to send the article for re-review after these edits, and think we can, in principle, offer acceptance at this time. kudos

Reviewer's Responses to Questions

**Comments to the Authors:**

Reviewer #1: The authors have successfully addressed the concerns of the initial reviewer reports with very helpful additions in the text, as well as extending the figures and tables with clarifying content.

Reviewer #2: Minor concerns

1. The authors addresses most of my concerns raised in the first round of review pretty well. Especially, it is glad to see Fig 1, Fig 2, Table 1 and Text Box involved. However, the parameters and rates like (including k, d, alpha, Pi) are still not involved in the diagram. Therefore, the coherent connectivity between the diagram of Michaelis Menten (Line 335) and the equations (Line 339-342) is not observed between Diagram (1.0) (Line 470) and Equations (1.1-2) (Line 472-3) of Model 1, between Diagram (2.0) (Line 501) and Equations (2.1-3) (Line 517-9) of Model 2, and between Diagram (3.0) (Line 534, 536-7, 543-4) and Equations (3.1-4) (Line 553-6) of Model 3. I highly recommend adding the model parameters and rates in the three diagrams.

2. This is not mentioned at the first round of review. However, I believe it is still worthy to discuss. Is the amount of R_{i free} in Ri_Ri is one or two? If it is two, Ri_Ri must be subtracted from Ri twice (through the two sigma notations), and the last term, +Ri_Ri, must be dropped.

3. Section numbers: 3.1 -> 2.1, 3.2 -> 2.2, ..., 4.1 -> 3.1, ...

4. Line 218: "in in" -> "in"

5. Line 335: "E-S" -> "E_S"

6. Line 394: "P" -> "P ij"

7. Line 394: "R_R=[R1_R2,R1_R2,..." -> "R_R=[R1_R1,R1_R2,..."

Reviewer #3: I have no further comments.

**Have the authors made all data and (if applicable) computational code underlying the findings in their manuscript fully available?**

Reviewer #1: Yes

Reviewer #2: Yes

Reviewer #3: None

PLOS authors have the option to publish the peer review history of their article (what does this mean?). If published, this will include your full peer review and any attached files.

Reviewer #1: No

Reviewer #2: No

Reviewer #3: No

---

## [Editor Report · Acceptance letter]

19 Dec 2024

PCOMPBIOL-D-24-00812R1 

Biomathematical enzyme kinetics model of prebiotic autocatalytic RNA networks: degenerating parasite-specific hyperparasite catalysts confer parasite resistance and herald the birth of molecular immunity

Dear Dr Conrad,

I am pleased to inform you that your manuscript has been formally accepted for publication in PLOS Computational Biology. Your manuscript is now with our production department and you will be notified of the publication date in due course.

With kind regards,

Lilla Horvath
